# C57bl/6 Mice Show Equivalent Taste Preferences toward Ruminant and Industrial Trans Fatty Acids

**DOI:** 10.3390/nu15030610

**Published:** 2023-01-24

**Authors:** Farzad Mohammadi, Nicolas Bertrand, Iwona Rudkowska

**Affiliations:** 1Endocrinology and Nephrology Unit, CHU de Québec-Université Laval Research Center, Québec City, QC G1V 4G2, Canada; 2Département de Kinésiologie, Université Laval, Québec City, QC G1V 0A6, Canada; 3Faculté de Pharmacie, Université Laval, Québec City, QC G1V0A6, Canada

**Keywords:** ruminant trans fatty acids, industrial trans fatty acids, encapsulation, IntelliCage, fat preference

## Abstract

Two distinct types of trans fatty acids (TFA) are found in the diet. Industrial TFA such as elaidic acid (EA) have deleterious effects on metabolic risk factors, and oppositely ruminant TFA including trans-palmitoleic acid (TPA) may have beneficial effects. The objective is to evaluate the taste preference between EA, TPA, lecithin or water. In this study, 24 female C57BL/6 mice were microchipped and placed in two separate IntelliCages^®^. Nano encapsulated TFA or lecithin were added to drinking water in different corners of the cage with normal diet. The study was carried out over 5 weeks, during which mice were exposed to water only (weeks 1 and 3), TFA or lecithin (week 2), and EA or TPA (weeks 4 and 5). Mice weights, corner visits, nose pokes (NP), and lick number were measured each week. The results demonstrated that mice consume more TFA, either EA or TPA, compared with lecithin. In addition, the mice licked more EA compared with TPA in one cage; conversely, in the other cage they licked more TPA compared with EA. However, when TFA positions were swapped, mice had equal licks for EA and TPA. In sum, mice preferred TFA, in equal matter compared with controls; therefore, the results demonstrate the potential for TFA-type substitution in diet.

## 1. Introduction

Fat is the most concentrated source of dietary energy, and also contributes to the texture, flavor, and aroma of a wide variety of foods [1]. The taste, smell, mouthfeel, and hedonic properties of fat all contribute to the popular concept of fat “taste” [2]. Fat is one reason why palatability and energy density of foods are closely intertwined [3]. Energy-dense foods and diets have been associated with higher energy intakes and with higher prevalence of diseases such as cardiovascular diseases, inflammation, and type 2 diabetes (T2D) [4,5,6]. Total fat intake has been reported to constitute 29.6–32.9% (66–73 g/2000 Kcal diet) of the typical diet in Canada [7] and fat intake recommendations are in the range of 20–35% of total energy intake (44–78 gr /2000 Kcal diet) [8]. However, the associations between different dietary fats and metabolic health still seem to be conflicting. Further, food preference can be altered by the type of fats in the diet [9]. Two different types of trans fatty acids (TFA) are found in the diet with potential opposite effects on metabolic health.

TFA are a group of fatty acids with at least one double bond in the trans configuration [10]. There are two main sources of dietary TFAs: TFAs resulting from the industrial partial hydrogenation of edible oils (I-TFA) and TFAs originating from bacterial hydrogenation of unsaturated fatty acids by ruminants (R-TFA). I-TFA include elaidic acid (EA; shorthand notation t9-18:1, t-18:1n-9 or Δ-nomenclature: C18:1 ∆9t) which is the trans isomer of oleic acid [11] and can be present in products such as margarine, snacks, and baked and fried foods. There are numerous studies showing the harmful effects of I-TFA [12,13]. Consumption of I-TFA have adverse effects on insulin resistance, lipid metabolism, and inflammation [14,15]. In contrast, R-TFA mainly consist of trans-vaccenic acid (TVA) with the trans double bond in position n-7 (shorthand notation t11-18:1, t-18:1n-7 or Δ-nomenclature: C18:1 ∆11t) and trans-palmitoleic acid (TPA) with one trans double bond in the n-7 position considering the Ω-nomenclature (trans-C16:1 n-7) or in position 9 considering the Δ-nomenclature (trans-9-C16:1) [16]. R-TFA are mainly found in dairy and meat products [17]. The intake of R-TFA may have health benefits by suppressing inflammatory pathways [18], lowering insulin resistance, and preventing atherogenic dyslipidemia [14]. However, the studies are scarcer. TFA consumption, mainly I-TFA consumption, has been reported to constitute 1.4% of the total energy intake (0.2–6.5%) worldwide [19]. R-TFA is expected to be <0.5% of the total energy intake [20]. Therefore, the substitution of I-TFA with R-TFA in the daily diet might provide beneficial health effects. In the context of bioactive molecules, a slight change in molecular structure may have impact on the biological properties. For example, changing the position of trans double bond in TFAs has implications for the perception, absorption, and break down of fat [10]. However, the exact mechanism is still poorly understood, and future investigations are warranted.

The palatability of food results from the oral perception of textures as well as chemical senses of taste and olfaction [3]. The sensory properties of fat taste seem to be perceived through a combination of olfactory and non-olfactory cues [9,21]. The first sensation is the aroma (smell or odor) of fat-soluble volatile flavor molecules, perceived through the nose or mouth. Later, oral sensations involve the texture of foods as it changes with time, from first bite to chewing and swallowing [22,23]. Direct biochemical sensing of fat molecules in the oral cavity and the gastro-intestinal tract also appears to be involved in the taste inclination toward fat-containing foods [22]. All combined phenomena result in a higher preference for fat in mice [24].

Animal taste preferences are assessed through evaluating the relative acceptance of different foods or food ingredients [25]. Various methods such as two-bottle preference test [26] and 2-pan choice method [27] have been used to assess food preference in animals. Due to the complexity of assessing the oral intake of solid diet, which involves individual housing of animals and frequent weighing of mangers [28], the introduction of fatty acids in the form of solution was proposed [23]. Recently, IntelliCage^®^ was introduced as a novel method to study animal’s preference toward foods. IntelliCage^®^ is a fully automated cage system designed to study behaviors of rodents living in a well-simulated social environment [23,29]. In our own previous work, we showed that lecithin vesicles encapsulating TFAs could be used in IntelliCage^®^ to investigate the taste preferences of mice. In that context, mice appeared to prefer I-TFA encapsulated in lecithin vesicles more than control vesicles [23]. However, this past study did not evaluate whether this preference was also observed toward R-TFA.

The main hypothesis of the study is that animals would prefer to consume TFA over lecithin and/or water. The secondary hypothesis is that animals would prefer EA over TPA. Thus, the objective was to evaluate the taste preference between I-TFA and R-TFA using the automated IntelliCage^®^ system in C57BL/6 mice, using lecithin vesicles and water as controls. This study examined how different TFAs can influence taste perception, and potentially paves the way to curb certain metabolic risk factors by substituting the type of fatty acids in food.

## 2. Materials and Methods

### 2.1. Materials

Commercial-adjusted calorie-purified Teklad Global 18% Protein Rodent Diet (58% calorie from carbohydrate, 24% protein and 18% from fat) was purchased from Envigo International Holdings, Inc. The complete information of the diet is available online (www.envigo.com, product 2018S, accessed online 17 January 2023). EA and TPA in the form of free fatty acids (purity > 99%, Product no.: U-47-A, U49-A, U-41-A, respectively) were purchased from Nu-check Prep, Inc. (Elysian, MN, USA). Soy lecithin Ultralec^®^ F (97%, Product no: 2516, CAS# 8030-76-0) was donated by Medisca, Inc (Montréal, QC, Canada). All other chemicals used in this study were purchased from Sigma-Aldrich (St. Louis, MO, USA) or Fisher Scientific (Waltham, MA, USA).

### 2.2. Nanovesicle Preparation

Lecithin nanovesicles were prepared using a solvent-free hydration method recently developed [30]. Briefly, oxygen was removed from ultrapure water by bubbling nitrogen. Lecithin powder (with or without 14 wt.% TPA and EA) was added and hydrated to reach to a concentration of 20 mg/mL. The 86:14 wt.% ratio for lecithin and TFAs was chosen based on previous experience [23], as it allows efficient encapsulation of the fatty acid in the phospholipids. This ratio provides a TFA intake that is representative of human consumption (1.1 to 1.6% of total energy intake as TFA with 6.5 mL of liquid consumed daily [31]). Hydration was conducted under magnetic stirring at 60 °C for 6 h. Liposomes were extruded using LiposoFast^®^ LF-50 medium pressure extruder (Avestin, Ottawa, ON, Canada) to produce unilamellar vesicles. Polycarbonate membranes with pores of 400, 200, and 100 nm were used sequentially. The solutions were prepared 1–3 days before the experiment each week and kept in a fridge (4° C). The composition of fatty acids in vesicle formulations was measured 3 days (the day before experiment) and 10 days (at the end of each week) after preparation by gas chromatography as previously described [23,32]. The average diameter (Z-average) and polydispersity index (PDI) of nanovesicles were determined by dynamic light scattering, on a Malvern Zetasizer Nano S instrument. The measurements were performed at a temperature of 22 °C and a backscatter angle of 173°.

### 2.3. Animal Study

The experiment was carried out in accordance with the guidelines of the Canadian Council on Animal Care (CCAP: standards and animal research: reporting in vivo experiments guidelines) and approved by Université Laval (Protocol: 2021-836, CHU-21-836). Twenty-four female C57BL/6 healthy animals were purchased from Charles River (Canada) and accommodated in a controlled environment (22 °C, 12 h day/night cycle) with ad libitum access to food and water (except during behavioral studies, where water was replaced by aqueous suspensions containing vesicles).

Twenty-four female C57BL/6 mice were acclimated for 72 h and microchipped by surgical insertion of subcutaneous radio-frequency identification (RFID)-transponders (ID 162-PM, ISO-compliant Transponder ISO 11784/11785, Peddy-Mark, Essex, United Kingdom). Animals had a one-week recovery and then were housed in two groups of 12 animals, in two IntelliCage^®^ systems (New Behavior AG, Zürich, Switzerland). Each IntelliCage^®^ (20.5 cm high, 55 cm × 37.5 cm at the base, Model 2000 Tecniplast, Buguggiate, VA, Italy) consists of four operant corners and each corner has two bottles (total of 8 bottles). Animals had access to each corner through a front hole. When animals entered a corner, a special RFID antenna recorded data on the number of animals present (corner visit) and the duration of each visit. Once animals were in a corner, nose poking was required for them to reach the bottles. Dedicated sensors in each corner were used to detect the number, duration, and side (right or left) of NP. The amount of consumed solution was measured by a lickometer that recorded the number and duration of licks. Shelters were provided in the center of the cage to serve as sleeping areas and stands to reach the food.

Throughout week 1–4 of the present experiment, only one bottle was used in each corner; during week #5, each corner also contained one bottle of water, in addition to the tested TFA-containing vesicles. Upon introduction to the cages, the animals had 3 days of acclimation before the start of the experiment. During week #1, both groups of mice had access to only water (control). At week #2, mice were given blank lecithin vesicles and either EA (cage 1) or TPA nanovesicles, (cage 2). Week #3 was a washout period where animals had only access to water (nanovesicles were removed). For week #4, all animals could choose between EA or TPA. The TFA used during week #2 (EA for cage 1, and TPA for cage 2) was reintroduced to the same corners, while the alternative TFA was placed in the other corners. During week #5, the positions of the TFA were interchanged to minimize bias due to site preferences (i.e., animals prefer one corner to another), and water was introduced to all corners. When vesicles were present (weeks #2, #4, and #5), two bottles of each formulation were available and placed in diagonal positions. Animals had free access to food during the experiment (Figure 1).

### 2.4. Statistical Analyses

Data were analyzed using GraphPad Prism 7 and SPSS software (version 18, SPSS Inc., Chicago, IL, USA). Repeated measures ANOVA was used to determine the differences in body weight. An unpaired *t*-test was carried out to assess the difference in daily number of licks, NP, and corner visits during the study. Simple linear regression was used to test if the number of the licks change during days in each week for corners 1 and 3 and 2 and 4. Throughout, *p* values ≤ 0.05 were considered as statistically significant.

## 3. Results

### 3.1. Vesicles Contain Similar Amounts of TFA and Remain Physicaly and Chemically Stable over a Period of 10 Days

Figure 2a shows the size distribution of vesicles over a 10 day period. These vesicles were approximately 200 nm in diameter at the beginning and end of the ten days, showing that the vesicles are physically stable for a period of up to ten days. Figure 2b shows the fatty acid composition of the encapsulated TFA three and ten days after preparation. The vesicles consist of linoleic acid, palmitic acid, elaidic acid, palmitelaidic acid, oleic acid, alpha-linolenic acid, vaccenic acid and stearic acids. Encapsulated TFAs were comparable three and ten days after preparation, for both EA and TPA formulations. Moreover, TFA contents were comparable between formulations, as the concentration of encapsulated EA and TPA were 17.1 and 17.7 wt%, respectively. Based on normal fluid consumption of mice (6.5 mL per day [31]), a 20 mg/mL formulation would result in physiological intake of TFA (i.e., 1.1 to 1.6% of the total energy).

### 3.2. Animals Gained Weight Consistently during the Study, and Weight Gain Was Comparable in Both Cages

Figure 3a shows the weight of animals during the study. Repeated measurements with ANOVA showed that the body weight of the animals did not differ between the two cages (*p*-value > 0.05). In addition, after 5 weeks, the animals gained 2.65 g in both cages (*p*-value < 0.05), i.e., about 15% of their initial weight.

### 3.3. IntelliCage^®^ Allow to Document the Preference of Mice for TFA-Containing Vesicles

The total number of licks over a period (i.e., the total amount consumed) provides information about the animals’ preference for specific formulations. This is similar to two-bottle experiments that collect data for individual animals to compare preference. However, IntelliCage^®^ provides complementary information that allows for dynamic experiments where animal choices change over time. Potential variables that could bias interpretation of results can be identified, and if animals have developed certain habits (e.g., a preference for certain corners), daily behavioral changes can highlight preferences that would go unnoticed in two-bottle experiments.

In a first analysis, we investigated whether the number of licks needed to be normalized for the body weight of the individuals. This evaluation was justified by previous observations in which larger mice consumed more fluid than smaller animals [31]. In our case, a linear regression analysis between the weight of the animals and the total number of their licks showed no relationship between the two parameters (Figure 3b). Therefore, we decided not to perform further analyses based on the weight of the animals.

Because the formulations were placed in different corners of the cage, we had to verify that all locations remained accessible to the animals throughout the study. Sociable behavior could indirectly influence the number of licks, for example, if a mouse was harassed or discouraged from going to a particular corner by other mice. Figure 4 shows that the activity of the mice (as measured by the number of nose pokes) remained constant over 5 weeks. In both cages and regardless of the formulations included, the mice elicited nose pokes in each corner approximately 100–200 times per day. This observation confirms that the mice had access to all formulations throughout the study.

For the same reasons, gregarious behaviors could also influence the number of licks. Hence, we evaluated if the number of licks had to be normalized by the number of nosepokes (number of licks/ nosepokes). We found a good correlation between the number of licks and the licks/nosepokes ratio (Appendix A), suggesting that no normalization was necessary, and that the number of licks was a good indicator of fat preference. The exact amount of solution consumption for each corner, in both cages, is shown in Table 1.

In the first week, animals in cage 1 licked similarly often (1.2 times more in corners 1 and 3 (167 mL) compared to corners 2 and 4 (135 mL) (*p* > 0.05) (Figure 5**)**. This lack of preference was to be expected since all corners contained water. In contrast, animals from cage 2 drank 1.9 times more from corners 1 and 3 (186 mL) compared with corners 2 and 4 (136 mL) (*p* < 0.05) during the same period (Figure 5). Since the formulations were identical, this difference was attributed to a topographic preference for corners 1 and 3, possibly due to unrelated external factors. To further investigate these unanticipated habits of animals in cage 2, we looked at the daily changes in the behavior of the mice during the control week. Figure 6 shows how the daily number of lickings that fluctuates during each week. Table 1 also shows the amount solution consumption in each week for different corners. During week 1, animals in both cages maintained a relatively constant number of licks between all corners. Although animals from cage 2 slightly favored corners 1 and 3 at the onset of the experiment (approximately 1.5-fold higher lick numbers compared with corners 2 and 4, on day 1), this preference remained relatively constant during the week.

During week 2, animals had 7.5 times more licks from EA compared with lecithin, in cage 1 (210 mL vs. 52 mL, *p* < 0.05), while mice from cage 2 had 8.6 times more licks for TPA compared with blank lecithin (206 vs. 82 mL, *p* < 0.05). The increase in the total number of licks observed in TFA-containing corners in both cages can be ascribed to a preference for the fatty acid containing vesicles. Further, it was also found that in week 2 the number of licks increased significantly during the week for EA and TPA compared with lecithin in both cages (*p* > 0.05). These findings were confirmed with the ratio of lick/nosepokes at week 2 (*p* > 0.05).

In week 3, when all vesicle formulations were replaced by water, animals from cage 1 maintained their preference for corners 1 and 3 (182 mL) compared to corners 2 and 4 (53 mL) by 4.7-fold more licks (*p* < 0.05). In cage 2, animals had 3.4 times more licks from corners 1 and 3 (176 mL) compared with corners 2 and 4 (92 mL) (*p* < 0.05).

In week 4, the original TFA was reintroduced to corners 1 and 3 (i.e., EA in cage 1 and TPA in cage 2), but animals could access the other type of TFA in corners 2 and 4. Like before, animals from cage 1 maintained their preference for EA (232 mL) compared to TPA (76 mL) by 3.1 times more licks (*p* < 0.05). Despite the lower number of licks for TPA, the number of the licks were increasing day to day over the course of the week (*p* < 0.05). Interestingly, animals from cage 2 also maintained their preference for TPA (196 mL) compared to EA (142 mL) by 1.3 times more licks (*p* > 0.05). However, it was found that the number of EA licks increased significantly from day to day during the week (*p* > 0.05).

In week 5, the locations of EA and TPA were switched, and water was introduced. In cage 1, the 2.6-fold higher number of licks of TPA (152 mL) compared with EA (54 mL) was not significant (*p* > 0.05). Similarly in cage 2, the animals had 1.4-fold more licks for TPA (136 mL) compared with EA (95 mL), this difference was not significant either (*p* > 0.05). Likewise, no significant changes were observed in the number of the daily licks for either cage. In cage 1, animals drank 1.4-fold more water from corners 1 and 3 than from corners 2 and 4 (38 vs. 32 mL, *p* < 0.05). In contrast, in cage 2, animals drank more water from corner 2 and 4 compared with corner 1 and 3 (1.7-fold more licks, 50 vs. 32 mL).

## 4. Discussion

Comparable formulations containing EA and TPA were prepared, and their TFA content remained stable throughout the study. In our study, all vesicles were extruded through polycarbonate membranes to obtain a diameter of about 200 nm. Reducing the size made the formulations more homogeneous and increased the physical stability of the formulations. The size of the droplets could play a role in animals’ preference for fats. In a study [32], the effect of droplet size of oil (triacylglycerol (a dietary fat in form of three fatty acids esterified to a glycerol molecule) purified from tuna, sardine and soybean oils) was investigated using two bottle choice tests in olfactory-blocked mice. In this study animals were administered emulsion with droplet size of 1 μm and 5.5 μm. The results showed that emulsion with 1 μm droplets were preferred up to 2 times more over emulsion with 5.5 μm droplets. However, the authors did not propose any mechanism to explain why smaller droplets were preferred over larger ones. Herein, comparable sizes between formulations ensured that preferences were not due to differences in vesicle diameter.

Herein, female C57BL/6 mice preferred TFAs vesicles were compared to blank lecithin vesicles. Exposure of animals to the formulations resulted in similar weight gain, in both cages. This is in accordance with our previous findings [23]. The present study was not designed to determine whether this preference could be explained by a change in taste or other organoleptic properties, such as odor and texture. Studies have shown that mice and rats with partial or complete loss of sense of smell because of blocked olfactory still have a spontaneous attraction to fats [9,21]. For example, when mice were given vegetable oils and xanthan gum with similar texture to vegetable oils, the mice continued to prefer vegetable oil [33]. Thus, taste appears to play an important role in preference for fat.

In the present study, we used TFA in their free acid form (99% purity). In another study [9], rats were given oleate (a fatty acid that occurs naturally in various animal and vegetable fats and oils) and triolein (a triglyceride) dissolved in 0.3% xanthan gum using a two-bottle preference test. The results showed that the animals preferred oleate to triolein [9]. Receptors on the tongue of animals such as free fatty acid receptor 1 (FFAR1), also known as G protein-coupled receptor 40 (GPR40), FFAR4 (GPR120), and CD36 recognize free fatty acids and not esters such as triglycerides and phospholipids [34]. These receptors are expressed in the taste epithelium of the mouse tongue and relay information about fatty acids to the brain [35,36]. Thus, activation of lingual fatty acid receptors could explain the preference of animals toward TFA.

C57BL/6J mice tend to consume less lecithin than other strains (129Sv/ImJ, and C57129F1 hybrids) [37]. While the intrinsically bitter taste of lecithin could also explain our observations, this is unlikely because the phospholipid was present in all formulations. Furthermore, through an unknown underlying mechanism, mice can develop tolerance to the bitter taste, following long-term exposure [38].

Our study concludes that mice showed comparable preferences toward EA and TPA vesicles. This is in accordance with another study that conducted two-bottle choice tests with corn, canola, and mixed vegetable oils (mostly triglycerides, at a concentration of 1% in the emulsion) [33]. In that setting, normal mice preferred vegetable oils to a control fluid, irrespective of the composition [33]. Another study [39] showed that the number of unsaturated carbons alter taste preferences in rats, who favored intake of fatty acids with larger numbers of double bonds (linolenic acid (18:3) > linoleic acid (18:2) > oleic acid (18:1)). Our study is the first to show that the taste preference for TFA with equal numbers of unsaturated carbons are similar. Comparisons with other fatty acids should be examined if future studies.

The present study was conducted in animals with normal diets (18% fat in diet). Results add to the current literature that rodents, like humans, display preferences for fat-rich foods [40]. Similarly, a study showed *ob*/*ob* mice preferred a high-fat diet (HFD, carbohydrate 20 kcal%; protein 20 kcal%; fat 60 kcal%) when compared with control diet (carbohydrate 70 kcal%; protein 20 kcal%; fat 10 kcal%) [41]. Another study also demonstrated a preference for high fat intake in progranulin knockout mice (Grn−/−), where a preference for 2% milk over 0.3% milk was observed [24]. This increased consumption of energy resulted in 10% body weight gain, over a period of 7 days. Interestingly, control mice did not demonstrate preference toward fatter milk, and their body weight did not change [24]. Preference toward fat could be related to dopamine-related pathways [42]. In this study rats were given either water or corn oil for 20 min after overnight food and water deprivation. The results showed dopamine increased after corn oil intake, but not water [42]. The higher dopamine signals could also lead to an addiction-like behavior in which licking more fat results in higher urge to drink more fat [43]. However, further studies need to confirm whether fat preferences are greater in obese mice compared to normal mice.

This study has two major strengths: first, the IntelliCage^®^ simulates the living conditions of animals in their social environment, while data on individual behaviour are still accessible. In addition, the social life of the animals reduces fear, stress, and anxiety, during the cage. The automated cage setup enables long-term, continuous cages with minimal human interference and high reproducibility. Second, the physiological dose of given TFA (1.1 to 1.6% of total energy intake) which corresponds to realistic dietary intake. In humans, the global consumption of TFA represents 1.4% of the total energy intake of which mostly is from I-TFA, while consumption of R-TFA is about <0.5% of the total energy intake [19,20]. R-TFA represents 2 to 5% of total fatty acids in dairy products and 3 to 9% in beef and lamb [44].

One limitation of this study is that all mice were female to avoid fights between individuals. Housing in large groups was deemed essential to draw conclusions despite inter-individual variability in behavior, as observed elsewhere [45]. While this limits the generalizability of the study, traditional two-bottle studies could be implemented; to test taste preferences in males. Likewise, in future studies, it might be valuable to extend the periods of run-in (week 1) and washout (week 3), to fully grasp the behavior of animals “at equilibrium”. In certain circumstances, introducing new formulations did not always fully alter the behavior of mice (e.g., in week 3, mice kept preferring some corners, despite that all formulations were equivalent). In a different setting, a “memory” for a preferred corner was also observed, even after mice spent 72 h outside of the IntelliCage^®^ [46]. In the present study, monitoring the daily changes in behaviors proved beneficial to understand how animals adapted to new formulations over time. Introducing new objects to the habitat can result in increased curiosity and corner visits [47]. However, herein, constant numbers of corner visits (nosepokes) strengthen our confidence that observations are due to taste preferences.

## 5. Conclusions

Overall, the mice preferred fat over water and TFA over lecithin. The preference for fatty acids may be related to the presence of fatty acid receptors in the tongue of the animals, which could explain the same preference toward both TFAs. Considering the potentially beneficial effects of TPA (R-TFA) compared with EA (I-TFA), future studies are needed to explore the mechanisms associated with the effects of these TFAs to be able to substitute EA with TPA intake. Ultimately, human taste studies need to be conducted to investigate the sensory evaluation of different TFAs.

## Figures and Tables

**Figure 1 nutrients-15-00610-f001:**
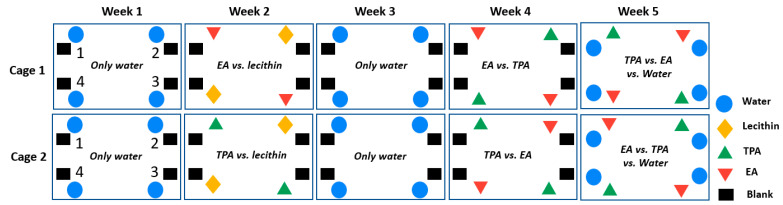
Experimental procedure of animal study during 5 weeks in 2 cages, W: week. Blue circle represents water bottle, yellow diamond represents lecithin vesicles, green triangle represents TPA formulation, red triangle represents EA formulations and black rectangle represents blank bottle for each corner.

**Figure 2 nutrients-15-00610-f002:**
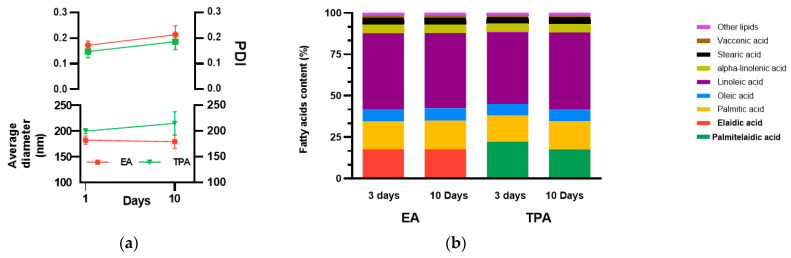
(**a**) Size distribution of vesicles. The vesicles are physically stable for periods up to 10 days; (**b**) composition of TFAs in nanovesicles. The vesicles were chemically stable for periods up to 10 days.

**Figure 3 nutrients-15-00610-f003:**
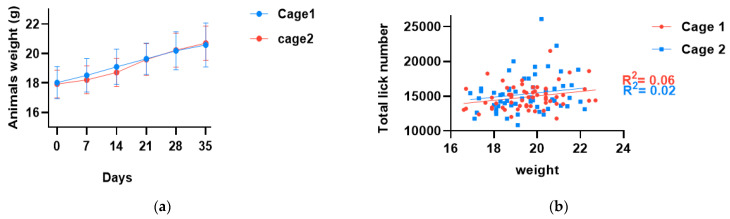
(**a**) Change in weight of animals during the study. Data represent means ± SD. After 5 weeks, animals gained 2.65 g in both experiments (*p*-value < 0.05); (**b**) number of licks normalized for the body weight of individuals; linear regression analysis shows no relation between weight and total number of the licks.

**Figure 4 nutrients-15-00610-f004:**
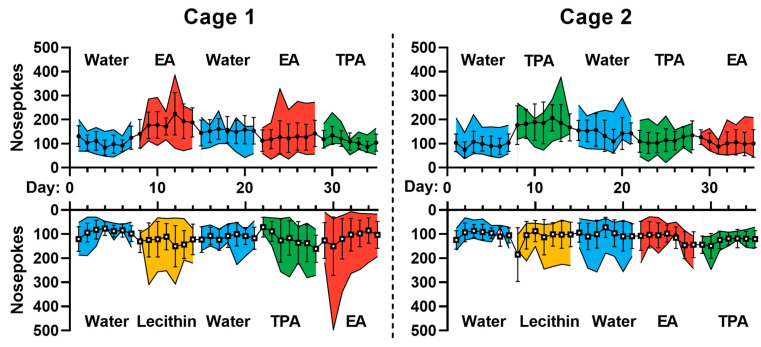
The activity of the mice based on the number of nose pokes. Data represent means ± SD. For each cage the graph on the top represents corner 1 and 3 and the one at the bottom represents corner 2 and 4. The activity of mice remains constant over 5 weeks as mice had approximately 100–200 nosepokes per day, in each corner.

**Figure 5 nutrients-15-00610-f005:**
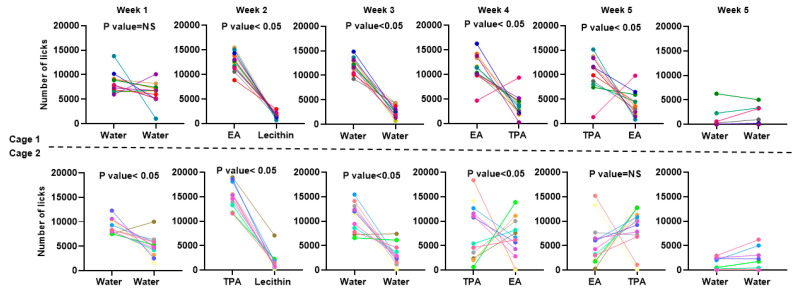
Weekly number of the licks and consumption of solutions in cage 1. Each box shows the number of the lick for 1 week; each point with a specific color represents an individual mouse. The differences between the lick numbers were evaluated by unpaired *t*-test.

**Figure 6 nutrients-15-00610-f006:**
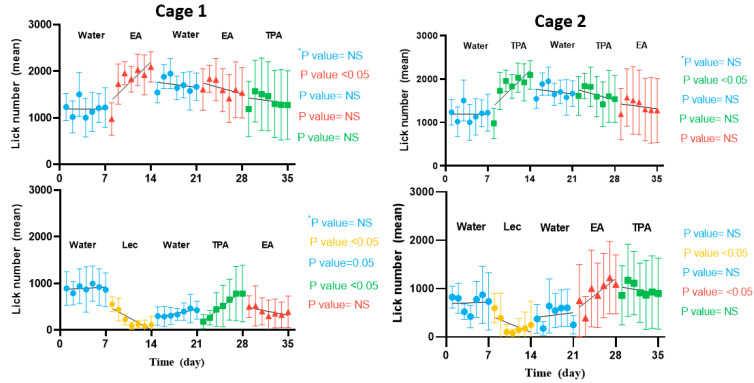
Changes in daily behavior of the animals during each week in both cages. Data represent means ± SD. Each bar shows the regression slope for each formulation. For each cage the box on the top represents corner 1 and 3 and the one at the bottom represents corner 2 and 4. * Significance of slope from zero.

**Table 1 nutrients-15-00610-t001:** Amount of solution consumption in each week for different corners (mL).

Cage 1	Corner 1	Corner 2	Corner 3	Corner 4
Week 1 (water)	94	44	73	91
Week 2	EA: 105	Lecithin: 22	EA: 105	Lecithin: 30
Week 3 (water)	78	22	104	31
Week 4	EA:120	TPA: 60	EA: 112	TPA: 16
Week 5	TPA: 106	EA: 12	TPA: 46	EA: 42
Week 5 (water)	18	12	20	20
Cage 2	Corner 1	Corner 2	Corner 3	Corner 4
Week 1	98	30	88	106
Week 2	TPA: 100	Lecithin: 62	TPA: 106	Lecithin: 20
Week 3(water)	47	38	129	54
Week 4	TPA: 82	EA: 58	TPA: 114	EA: 84
Week 5	EA: 52	TPA: 54	EA: 43	TPA: 82
Week 5 (water)	15	20	20	30

## Data Availability

Supporting data is available from the authors if requested.

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
