# Peer review of "C57bl/6 Mice Show Equivalent Taste Preferences toward Ruminant and Industrial Trans Fatty Acids"

_nutrients, 2023, doi:10.3390/nu15030610_

Round 1

Reviewer 1 Report

I have completed my evaluation of your manuscript entitled “C57bl/6 mice show equivalent taste preferences toward ruminant and industrial trans fatty acids”. Generally speaking, the manuscript was well-written and put forward valuable new findings. However, the article has a small amount of data and is not innovative enough. In fact, the results showing mice had a taste preference toward fats are not surprising, however, we may be more concerned about some functional differences caused by taste preferences for different fatty acids. Following were my questions and comments:

1.        In your study, why choose 14 wt.% TPA and EA to prepare nanovesicles, is this the best ratio or is there any basis for it?

2.        The article did not mention the method of determining the size distribution of vesicles.

3.        The “Animal study” section is hard to understand, so I suggest adding a photo (or concept drawing) of the IntelliCage® system, and adding a table (or figure) to show the exact experimental procedure to make it easier to read.

4.         The writing style of this article is hard to follow, the Introduction and Discussion sections of the article are redundant, so the content needs to be more concise. Besides, the presentation of figures and tables in the article is not standard, and the figures and tables must be improved.

Reviewer 2 Report

Aim of the present research is significant to the filed of food nutrition, and the content is reader-attracted. I suggest accept after revision.

1. The specific parameters method needs to be provided in Method Section;

2.  Add the discussion in Intro or Discussion Part on the difference of industrial TFA and natural TPA. Why? Reason? Structure?

3. The work of chemistry in this research is a little limited. The mechanism of the effect of different fatty acids intake on mice could be added.

4. The discussion in present status is more phenomenon description, the underlying mechamism is absent.

Round 2

Reviewer 1 Report

I agree it is accepted.